# Polyelectrolyte Encapsulation and Confinement within Protein Cage-Inspired Nanocompartments

**DOI:** 10.3390/pharmaceutics13101551

**Published:** 2021-09-24

**Authors:** Qing Liu, Ahmed Shaukat, Daniella Kyllönen, Mauri A. Kostiainen

**Affiliations:** 1Biohybrid Materials, Department of Bioproducts and Biosystems, Aalto University, 00076 Aalto, Finland; qing.2.liu@aalto.fi (Q.L.); ahmed.1.ahmed@aalto.fi (A.S.); daniella.kyllonen@aalto.fi (D.K.); 2HYBER Center, Department of Applied Physics, Aalto University, 00076 Aalto, Finland

**Keywords:** protein cage, polyelectrolyte, electrostatic interaction, nanocompartment, self-assembly, nanocoating

## Abstract

Protein cages are nanocompartments with a well-defined structure and monodisperse size. They are composed of several individual subunits and can be categorized as viral and non-viral protein cages. Native viral cages often exhibit a cationic interior, which binds the anionic nucleic acid genome through electrostatic interactions leading to efficient encapsulation. Non-viral cages can carry various cargo, ranging from small molecules to inorganic nanoparticles. Both cage types can be functionalized at targeted locations through genetic engineering or chemical modification to entrap materials through interactions that are inaccessible to wild-type cages. Moreover, the limited number of constitutional subunits ease the modification efforts, because a single modification on the subunit can lead to multiple functional sites on the cage surface. Increasing efforts have also been dedicated to the assembly of protein cage-mimicking structures or templated protein coatings. This review focuses on native and modified protein cages that have been used to encapsulate and package polyelectrolyte cargos and on the electrostatic interactions that are the driving force for the assembly of such structures. Selective encapsulation can protect the payload from the surroundings, shield the potential toxicity or even enhance the intended performance of the payload, which is appealing in drug or gene delivery and imaging.

## 1. Introduction

Precise compartmentalization plays a fundamental role in all life forms. While subcellular compartments developed in eukaryotes are comprised of lipids, likewise in prokaryotes, micro- or nano-compartments are dominantly protein-based [1,2]. In either case, precise compartmentalization leads to localized as well as concentrated sequentially acting enzymes, which boost the rate of catalytic cascades or sequesters the produced toxic substances [1,2,3,4].

Protein cages, such as ferritin, and viruses, such as cowpea chlorotic mottle virus (CCMV), are simple forms of nanocompartments found in nature. They are composed of multiple copies of one or a few unique subunits and adopt highly symmetric structures and uniform sizes, protecting or isolating entrapped payloads from external environment [5]. For example, mammalian ferritin has a spherical structure with an outer diameter of 12 nm and an inner cavity of 8 nm. The constitutional subunits contain a heavy subunit of 21 kDa and a light subunit of 19 kDa and the ratio of these two subunits depends on the tissue type [6]. Lumazine synthase from *Aquifex aeolicus*, on the other hand, is built from 60 identical subunits that assemble into icosahedral capsids with an outer diameter of 18 nm [7,8]. Native protein cages have evolved to display various distinct functions. While the ferritin regulates the availability of iron in almost all organisms, lumazine synthase is responsible for the production of riboflavin (vitamin B_2_) in, for example, fungi, yeasts, bacteria and plants [6,9]. Viral protein cages are loaded with nucleic acid genomes and are able to provide protection in varying environments and deliver genetic materials to host cells [5].

In the case of small RNA viruses, they are typically created by the complementary electrostatic interactions between cationic capsid proteins (CPs) and the anionic nucleic acid genome and are aided by non-covalent interactions between protein subunits [10]. Given the polyanionic nature of genetic materials, it can be expected that other polyanions can also template the capsid formation. The assembly/disassembly processes of viruses and virus-like particles (VLPs) remain elusive due to the inaccessibility of cargo–protein interactions and the atomic structures of transient assembly intermediates [11]. One of the many theories for capsid assembly is the nucleation and growth mechanism [12]. This involves the formation of a critical nucleus, which then goes on to a growth phase where one or few subunits combine until a capsid is completed. The critical nucleus can be defined as the smallest intermediate in the growth phase which has more than 50% probability to become a complete capsid before disassembling. Pertaining to icosahedral geometry, the initial intermediates are comparatively less stable because very few subunit–subunit interactions are present [12]. Owning to the geometrical shape of the critical nucleus i.e., polygon, the number of interactions are maximized as well as additional stabilization is achieved due to the conformation changes. This however depends on protein subunit concentration as well as solution conditions. Similar considerations apply to capsid disassembly. In order to have a large activation barrier, the first few subunits to disassemble must break many contacts [12].

Many studies have shown that the driving forces for the capsid assembly are hydrophobic interactions acting between apolar patches on the coat proteins [13,14] and there are many other factors that affect the assembly formation as well as the stability of the capsid. The factors notably include electrostatic interaction [13,14], pH [15], polymer length [16], and protein concentration [16]. The electrostatic interactions arise from the net electrical charge found on the coat proteins and hence are sensitive to ionic strength as well as the pH of the solution. Moreover, changes in ionic strength lead to conformation changes in the protein subunits, which can either activate or deactivate the assembly process [14]. In the case of CCMV, pH plays a crucial role in the capsid formation. Low ionic strength and neutral pH are required for the absorption of CP on the RNA. Subsequent lowering in pH increases the CP–CP interactions and favors the formation of an ordered capsid [17]. Therefore, optimizing these parameters is crucial for the formation of capsid assembly in vitro. This electrostatic-driven and cooperativity-aided assembly pathway has also inspired scientists to modulate native proteins to encapsulate extraneous payloads or to engineer protein cage-mimicking structures, in an attempt to enhance the cargo performance and to circumvent the drawbacks associated with viral cages [18]. For example, an engineered protein was found to encapsulate DNA individually, which transfected cells as efficiently as polyethyleneimine and Lipofectamine 2000 [19]. Surface coating of nanoparticles (NPs) with protein shells may enhance the cell uptake efficiency and avoid off-target accumulation and toxicity, leading to improved performance in biomedical fields [20]. Selected cages have endogenous pores, which can change their size in response to environmental cues and regulate how different materials can navigate in and out of their interiors. For example, anionic poly(anetholesulphonic acid) has been reported to load into an empty CCMV capsid at pH = 7.5 when the pores undergo a close-to-open transition, and a subsequent decrease in pH to below the gating threshold (pH = 4.5) can entrap the encapsulated polymers [21]. Unlike the native approach where the cargo encapsulation and capsid formation proceed simultaneously, the successful loading is performed with a preformed capsid and dominantly prompted by the electrostatic interaction, thus making an invaluable supplement for polyelectrolyte segregation.

While there are already some excellent reviews on protein cage hybrids and high-order lattices [5,10,18,22,23], this review specifically focuses on the advances in the core-shell assemblies formed by proteins (shell) and polyelectrolytes (core). First, the structures of native protein cages that have been intensively utilized to load extraneous polyelectrolytes are introduced. Next, the assemblies and applications of protein cages encapsulating extraneous polyelectrolytes through electrostatic interaction are addressed. Finally, the protein cage-resembling structures and coatings of proteins that are templated by polyelectrolytes through electrostatic interaction are discussed. It is also worth noting that the work covering encapsulation processes that are not electrostatically driven is not covered here. For instance, confined polymerization in the cavity of bacteriophage P22 was pioneered in Douglas’ group and the entrapped polymers were not restricted to charged ones [10].

## 2. Native Structures of Protein Cages

Protein cages occur in nature with various shapes and diverse functions, and they can be categorized into two families: non-viral protein cages (ferritin, heat-shock protein, etc.) and viral protein cages (CCMV, tobacco mosaic virus (TMV), etc.) [5]. Similar to non-viral protein cages, viral protein cages also adopt various morphologies and have cavity diameters ranging from ~30 nm (bacteriophage FX174) to over 500 nm (mimivirus) [10,18]. Many viral capsids consist of 60T protein subunits that adopt assemblies with icosahedral symmetry consisting of 12 capsid protein pentamers and a varying number (10(T-1)) of hexamers. The triangulation number (T number) can be used to describe the arrangement of the capsomers on a hexagonal grid, as proposed by the quasi-equivalence theory from Caspar and Klug [24,25]. To guarantee efficient genome encapsidation, viruses have naturally evolved with a cationic interior, which in turn provides an advantage in extraneous polyanion loading. Moreover, owing to the limited number of constitutional subunits, a point mutation at one subunit allows for the docking of multiple non-native functional groups around the protein cage following the symmetry of the cage. The above-mentioned properties have thus motivated researchers to target protein cages for desired applications. In this section, the native structures of the protein cages which have been intensively studied on the encapsulation of polyelectrolytes are introduced. They include brome mosaic virus (BMV), cowpea chlorotic mottle virus (CCMV), red clover necrotic mosaic virus (RCNMV) and lumazine synthase from *Aquifex aeolicus* (AaLS).

### 2.1. BMV

BMV is a *Poaceae*-infectious virus that consists of a protein capsid with an icosahedral symmetry and a positive-strand RNA genome [26]. The protein capsid contains 180 copies of identical subunits and has an outer diameter of 28 nm and an inner cavity of 18 nm (*T* = 3, Figure 1a). The capsid is formed by the interaction between the anionic genome and the positively charged CPs [26]. The BMV capsid can undergo a structural transition in response to pH: low pH (pH < 6.0) and magnesium ion (Mg^2+^) stabilize the BMV while neutral pH disassembles it into constitutional subunits [26]. Under the disassembly condition, the RNA genome can be precipitated by CaCl_2_, and empty capsid particles (VLPs) with different sizes (*T* = 1 and *T* = 3) can be obtained upon lowering the pH and ionic strength [27].

### 2.2. CCMV

CCMV specifically infects the cowpea plant and develops yellow spots on the leaves where the name “chlorotic” is derived. Like BMV, the coat protein forms an icosahedral capsid (*T* = 3) that packages a positive-strand RNA genome with an outer diameter of 28 nm and an inner cavity diameter of 18 nm (Figure 1b) [28]. The protein capsid is composed of 180 identical subunits and is stabilized by the polyanionic genome through electrostatic interaction [28]. Native CCMV is stable at pH = 5.0 but disassembles at pH = 7.5 or high ionic strength [28]. The assembly and disassembly processes are reversible simply by adjusting the pH or ionic strength [28]. Like BMV, after removing the genetic materials, empty CCMV VLPs can be obtained at pH = 5.0 [29]. Another important feature of CCMV is that the pores at the quasi-threefold axis undergo an open-close transition in response to pH or divalent cation (Ca^2+^ or Mg^2+^) concentration, which allows for a size-dependent entry and entrapment of materials [28].

### 2.3. RCNMV

RCNMV is a plant virus belonging to the *Dianthovirus* genus, *Tombusviridae* family. It contains two single-stranded RNA that consist of either 1 RNA-1 (4 kilo bases, 4kb) and 1 RNA-2 (1.5 kb) or 4 RNA-2 [30]. The RNA genome templates the assembly of 180 CPs into an icosahedral capsid with an outer diameter of 36 nm (*T* = 3) and an inner cavity of 17 nm (Figure 1c) [30,31]. The RCNMV virion contains divalent cations such as Ca^2+^ and Mg^2+^, and, similar to CCMV, the RCNMV capsid is switchable between an open and a close state through the control over the divalent cation concentration [31]. The divalent cations play a role in stabilizing the capsid structure and protecting the genomic RNA from degradation [31].

### 2.4. AaLS

Lumazine synthase (LS) is one of the bacterial enzymes that catalyze the biosynthesis of vitamin B_2_. While some LS exist as pentamers or dimers of the pentamer, lumazine synthase from *Aquifex aeolicus* (AaLS) is found to self-assemble into a cage-like structure, which consists of 60 copies of identical subunits and adopts an icosahedral symmetry with an outer diameter of 18 nm (*T* = 1, Figure 1d) [7]. The thermal stability of AaLS is high (*T_m_* = 119.9 °C) comparing to the LS counterparts from *Bacillus subtilis* (T*_m_* = 93.3 °C) and *Saccharomyces cerevisiae* (*T_m_* = 74.1 °C) [7].

## 3. Protein Cages Encapsulating Noncognate Polyelectrolytes

Given the polyanionic nature of a nucleic acid genome, it is reasonable to expect the efforts to replace the genetic materials with extraneous anionic polyelectrolytes. The well-defined cavity of protein cages serves as an ideal container for size-constrained material loading and synthesis. The precise structure, addressability and biocompatibility are also an advantage considering biomedical applications. There exist two main methods for polyelectrolyte encapsulation: (1) polyelectrolyte encapsulation during protein cage formation and (2) diffusion of polyelectrolytes into preformed protein cages. Below, studies on the polyelectrolyte encapsulation into the as-mentioned protein cages via the two methods are introduced.

### 3.1. BMV

The exact assembly process of native BMV remains unclear. One prevalent hypothesis suggests a process starting with the binding of the CP to the RNA genome through nonspecific electrostatic interaction [32]. The initially formed RNA–CP complex creates a nucleation point for the subsequent CP addition, and the CP–CP interaction promotes cooperative CP condensation around RNA, which ultimately leads to the virion formation. The CP–CP interaction is also proposed to be responsible for regulating the CP number for virus construction, and in the absence of the RNA genome, it is strong enough to direct the capsid assembly under low pH and ionic strength [27,32].

The earliest attempt to utilize extraneous polyelectrolytes to template the BMV VLP formation was reported in 1969 by Bracker et al. [33]. They discovered that it was the polyanionic phosphate backbone rather than the genomic sequence that nucleated the protein capsid formation. This assumption was supported by the observation of VLPs (CCMV, BMV and broad bean mottle virus) formed in the presence of a variety of homo- and hetero-poly(ribonucleotides) or two synthetic polyanions (polyvinyl sulfate and sodium dextran). Particularly, novel rod-like structures were observed when BMV CPs were mixed with a rigid calf thymus double-stranded DNA (dsDNA) (Figure 2a). The rod-like particles were structurally distinct from tubular CCMV capsids obtained in the absence of nucleating agents, which verifies the essential role of the template in determining the capsid structure [34].

The Dragnea group has focused on the encapsulation of inorganic NPs, such as gold nanoparticles (AuNPs) [35,36,37,38,39], iron oxide nanoparticles (IONPs) [40,41] and quantum dots (QDs) [42]. Initially, citrate or streptavidin functionalized AuNPs (2.5 and 4.5 nm) were employed to load BMV capsids [43]. However, only 2~3% capsids were successfully loaded. The loading efficiency was tremendously improved to 95 ± 5% when an anionic thiolalkylated tetraethylene glycol (TEG) was used to decorate 12 nm AuNPs (Figure 2b) [35]. The high loading yield was attributed to the cooperative CP binding mediated by the anionic particles with a proper size (16 ± 1.2 nm) that was similar to the native BMV cavity diameter (17~18 nm). Later studies demonstrated that it is feasible to manipulate the VLP morphology via adjusting the AuNP diameter [36]. BMV VLPs of *T* = 1, 2 and 3 were obtained when AuNPs of 6, 9 and 12 nm were used as the template, respectively, and the polymorphs were supported by a three-dimensional (3D) reconstruction (Figure 2c). Transmission electron microscopy (TEM) images revealed that the 12 nm AuNP had the highest loading efficiency, and the corresponding VLPs (VLP_12_) highly resembled the native *T* = 3 BMV (R3BMV). Co-crystallization of VLP_12_ and R3BMV yielded highly ordered crystals with a distinct optical absorption spectrum from that of VLP solutions. The changes in the optical spectrum are due to multipolar effects on the plasmon coupling, which can arise from both the crystal structure and the material composition, making such materials promising for tunable optical metamaterials. By changing the ratio between two ligands, an anionic TEG-COOH and a neutral TEG-OH, it was possible to adjust the surface charge density on AuNPs which was demonstrated to be critical for efficient VLP formation [38]. If the charge density was below a critical value, no capsid was observed even when the total charge was enough to build a closed protein capsid. This observation was explained by the low surface charge density that was not able to adsorb BMV CPs and create a nucleus for the subsequent capsid growth. It was also found that the surface charge density was proportional to VLP formation efficiency but had a limited effect on the size and morphology, implying a viable method to control the VLP integrity for potential applications in controlled drug release.

Recently, BMV spherocylindrical shells templated by gold nanorods were reported (Figure 2d) in an attempt to intentionally control the protein symmetry and ultimately find applications in molecular nanomaterials [37]. In situ atomic force microscopy (AFM) images showed that the CP binding followed the nanorod template and formed the spherocylindrical coating with icosahedral caps and a cylindrical side. Chirality and defect were observed and recapitulated by coarse-grained simulations. These observations were attributed to the competition between the intrinsic CP curvature and the inhomogeneous mean nanorod curvature. Experimental and simulation results offered insights into the BMV capsid growth mechanism and assembly pathway around polyhedral templates. Aside from AuNPs, IONPs with an anionic lipid micelle coating were also employed [40]. Though bigger than the cavity diameter, IONPs (20.1 nm) successfully led to a BMV VLP formation (41.3 nm) with a triangulation number beyond 3. The encapsulation efficiency derived from TEM results was 35 ± 5%. Similar values have been reported previously with 18 nm AuNPs [36]. However, a much lower assembly efficiency was found with smaller IONPs (~5% for 10.6 nm and ~3% for 8.5 nm). These values were also much lower than those of AuNPs with similar sizes (50%) [36]. The clear encapsulation efficiency difference was attributed to the surface modification-dependent average curvature and faceting effect of the NP core. Nevertheless, the superparamagnetic IONPs used in this work had a blocking temperature of ~250 K, which made the IONP-encapsulating VLPs appealing as magnetic resonance imaging agents and biomagnetic materials. Later, a series of poly(ethylene glycol) (PEG) chain grafted poly(maleic acid-alt-octadecene) polymers were utilized to coat IONPs (22~24 nm), and the obtained products were subsequently applied to template BMV and hepatitis B virus (HBV) VLP assemblies [41]. The large IONP size and the high coating molecular weight together yielded BMV VLPs of 42.8~48.1 nm and HBV VLPs of 39.9~43.1 nm. As indicated by TEM images, PEG chains in HBV VLPs were more likely to extend through the capsid pores while the tight BMV capsids constrained PEG chains in the chamber between the protein shell and the core, which accounted for the size difference between the two VLP types.

### 3.2. CCMV

CCMV is an icosahedral virus that can be reconstituted in vitro [44]. High-resolution reconstructions together with computational simulations have provided deep insights into the molecular structure and interaction within the virion [28,45]. However, like the structural analogue BMV, the assembly pathway remains poorly understood. A classic in vitro assembly mechanism was proposed by Zlotnick et al. [46,47]. Initially, CCMV CPs bind to RNA through electrostatic interactions and with low cooperativity. When the number of CP dimers on each RNA molecule reaches ten, RNA is neutralized and progressively folded into a compact morphology. Cooperative and stepwise binding of additional CPs to the nucleoprotein ultimately yields the intact virus. Similar to BMV, CCMV capsid formation is independent of the anionic genome.

Bracker et al. used polyribonucleotides and synthetic polyanions to template the CCMV VLP assembly [33]. Only spherical VLPs were observed even in the presence of a rigid calf thymus dsDNA. However, tubular VLPs were reported by Mukherjee et al. (Figure 3a) [48]. The length of the tubes was controllable by adjusting the stoichiometry of the DNA base pair (bp) and the CP dimer, but the tubular diameter remained constant (17 nm). Hemispherical caps were observed at the terminus of some tubes (marked by the white arrows in Figure 3a), indicating a half capsid (*T* = 1). The tube length reached as long as 5 μm, which was far longer than the DNA template. Taking into account the tubular diameter (~17 nm), it was suggested that several DNA duplexes aligned parallel along the tube axis and the continuous binding of CPs to the DNA stagger yielded the exceptional VLP structures. Ruiter et al. compared the cell uptake performances of the spherical and tubular VLPs [49]. It was found that both structures were internalized by the clathrin-mediated endocytosis pathway. VLPs localized predominantly on the membranes or in the endosomes and lysosomes after entering cells. Kwak et al. utilized DNA micelles as the template (Figure 3b) [50]. Through a modification on the nucleobase with a hydrophobic lipid chain and a subsequent introduction to a DNA sequence, an amphiphilic DNA conjugate was obtained, which was utilized to initiate the VLP formation (*T* = 1 and 2). The amphiphilic nature was highly feasible for the co-loading of hydrophobic drugs in the core and hydrophilic ones in the DNA corona through complementary DNA hybridization [51]. A similar system employing DNA corona to trigger CCMV capsid assembly was reported by Brasch et al. [52]. DNA chains were covalently attached to an enzyme and the DNA–enzyme hybrid led to the formation of icosahedral CCMV VLPs (*T* = 1), as evidenced by TEM images and a 3D cryo-EM reconstruction. Two-enzyme cascade was encapsulated in the VLPs, which were observed to be catalytically active, and may provide insights into the function of biological compartments. The CCMV CP assembly templated by a rectangular DNA origami nanostructure was significantly distinct from the native icosahedral symmetry (Figure 3c) [53]. It was found that initial CP binding occurred at a low CP to origami ratio and a gradual increase in CP ratio induced an origami shape transition from the original rectangle to a rolled tube. Further increase in CP concentration unwrapped the origami and led to a complete coverage on the origami. A cell transfection study revealed that increasing the CP coverage enhanced the transfection efficiency and, compared to the bare DNA origami, a maximum of a 13-fold enhancement in origami delivery was demonstrated.

Aside from nucleic acid-based materials and structures, synthetic anionic polyelectrolytes have been successfully employed to template CCMV VLP assemblies. Douglas et al. used a CCMV capsid as a reaction vessel for the controlled synthesis of two anionic polyoxometalate species, paratungstate and decavanadate [21]. The molecular tungstate (WO_4_^2−^) and vanadate (V_10_O_28_^6−^) were electrostatically attracted to the cavity under conditions where the VLPs were in a swollen state (pH > 6.5). Aggregation of the precursor monomers on the inner surface triggered the crystallization of the polyanionic polyoxometalate minerals. The obtained mineral crystals were entrapped in the cavity by closing the pores via lowering the pH (pH = 4.5). TEM results demonstrated the single-crystal nature of the mineral particles with a defined size and shape. They also proved the electrostatic aspect of the host–guest interaction by trapping an organic polyanion, (poly(anetholesulphonic acid)). Sikkema et al. reported the encapsulation of a flexible poly(styrene sulfonate) (PSS, 9.9 kDa) into CCMV VLPs with varying ratios, which produced monodisperse icosahedral particles (16 nm, *T* = 1) [29]. The anionic PSS was later found to stabilize CCMV VLPs (18 nm, *T* = 1) bearing an exterior PEG coating, a modification that provoked irreversible dissociation of native CCMV cages (Figure 4a) [54]. At neutral pH, wild-type CCMV was also found to disassemble into free CPs which reassembled around PSS polymers [55]. Isothermal titration calorimetry measurements revealed that the enthalpy change caused by the CP binding to PSS was six times higher compared to that induced by CP–ssDNA interaction, which contributed to the thermodynamically biased CCMV VLP formation around PSS. Chevreuil et al. used time-resolved small-angle X-ray scattering to study the assembly dynamics of the capsid around a PSS polymer (600 kDa) or an RNA genome [56]. Similar to the RNA genome, PSS rapidly captured CCMV subunits electrostatically and the resulting complexes slowly relaxed into spherical VLPs. The observation that CCMV VLPs assembled around PSS instead of RNA at neutral pH was explained by the flexibility and hydrophobicity of the organic polymer. Upon binding to CPs, PSS self-accumulated into a globular core through the interaction between the benzene rings, which allowed a tight CP–CP interaction and final formation of the closed capsid [57]. Hu et al. compared the VLP formation templated by PSS of five lengths to evaluate the competition effect between the protein subunit curvature and the enclosed polyelectrolyte size [58]. They found that the size of CCMV VLPs increased from 22 nm (*T* = 2) to 27 nm (*T* = 3) with a unimodal capsid size distribution for each PSS when the polymer weight was increased from 0.4 MDa to 3.4 MDa, implying the critical role of the encapsulated cargo size in determining the VLP size. They later reduced the PSS size to 38 kDa and found predominant 19 nm CCMV VLPs (*T* = 1) together with a small fraction of 21 nm particles (*T* = 2) at a low CP ratio [59]. An increase in the CP ratio reduced the 19 nm VLP to 70% of total particles. By employing rhodamine B functionalized PSS, it was unveiled that two PSS were in *T* = 1 capsids and three PSS in *T* = 2 capsids. Tolbert et al. studied the CCMV VLP assembly around a fluorescent semiconducting poly(2-methoxy-5-propyloxy sulfonate phenylene vinylene) (MPS-PPV), a polymer that could undergo an ionic strength-dependent conformation transition in aqueous solutions [60]. At low ionic strength (0 M NaCl), MPS-PPV adopted a stretched conformation, whereas it coiled at high ionic strength (1.0 M NaCl), and both conformations exhibited distinct luminescent properties. Together with the ability that the CP can adapt to the shape of the template, CCMV VLPs of tubular and spherical shapes were feasibly obtained by adjusting the ionic strength, as confirmed by TEM and fluorescence anisotropy (Figure 4b). Välimäki et al. conjugated a heparin-specific peptide (HBP) to the CP N-terminus with the SrtA-based method and approximately 25% of the CPs were successfully conjugated [61]. The addition of heparin triggered the CCMV VLP formation through charge–charge interactions between heparin and HBP. The assembly was biased over other glycosaminoglycan analogues and exhibited limited hemolytic activity, demonstrating a promising alternative for selective heparin-reversal.

Brasch et al. reported on the encapsulation of an anionic dye, zinc phthalocyanine (ZnPc), into CCMV VLPs (Figure 4c) [62]. The CPs were able to assemble around ZnPc stacks at neutral pH, forming 19 ± 5 nm VLPs (*T* = 1). ZnPc dimers could also diffuse into empty capsids (*T* = 3) at pH = 5, and no further ZnPc aggregation inside the capsid was observed. However, VLPs containing ZnPc dimers were not stable as they switched into *T* = 1 VLPs and ZnPc dimers reengaged into stacks inside the capsid when the pH was elevated to 7.5, demonstrating the dominating roles of the electrostatic force and the cargo size in VLP stabilization. Irradiation of macrophage cells incubated with ZnPc-loaded VLPs provoked the death of most cells (bottom panel, Figure 4c), demonstrating the potential in photo-dynamic therapy. Further studies revealed that the ZnPc stacks in VLPs had a uniform spherical nanostructure (10 nm), a synergistic effect of the symmetry and size constraint imposed by the capsid [64]. The structure and stability of the resultant VLPs were also influenced by the ZnPc stacks. Later, a library of Pc dendrimers was also synthesized to template the CCMV VLP assembly [65]. ZnPc dendrimers contained four Fréchet-type dendritic wedges at the periphery while ruthenium Pc (RuPc) dendrimers were substituted axially. It was found the zero (ZnPc1 and RuPc1) and the first (ZnPc2 and RuPc2) generations successfully induced the VLP formation (18 nm, *T* = 1) but full capsid formation was only observed with ZnPc1 and RuPc2. Millán et al. loaded ZnPc to a paramagnetic micellar structure formed by 1,4,7,10-tetraaza-1-(1-carboxymethylundecane)-4,7,10-triacetic acid cyclododecane and GdIII complex (Gd-DOTAC10) [66]. The introduction of ZnPc to the micellar core stabilized the micelles through hydrophobic interaction and favored the micelle encapsulation within the capsid. As a result, the paramagnetic VLPs exhibited enhanced r_1_ relaxivity compared to Gd-DOTAC10 in solution, demonstrated by the significantly higher contrast in magnetic resonance images. Sinn et al. reported the CCMV VLP formation that was triggered by negatively charged luminescent platinum (Pt(II)) complex amphiphiles (Figure 4d) [63]. The structures of the Pt(II) amphiphiles were delicately designed, yielding Pt(II) complexes with two morphologies (sphere and tube) that exhibited outstanding aggregation-induced luminescence. These two structures were found to template icosahedral and rod-like VLP assemblies, respectively. Moreover, both VLPs showed exceptional quantum yields (51~57% for icosahedral VLPs and 23% for rod-like VLPs), and they were visible at room temperature. The unique photophysical properties of the enclosed chromophores (Pc or Pt(II) complexes) thus made the as described CCMV VLPs promising candidates in future imaging applications.

Polyanionic inorganic NPs present another important category in CCMV VLP encapsulation. Aniagyei et al. used a rigid polyanionic template, TEG-coated AuNP (11.6 ± 0.9 nm), to template the assembly of a CCMV mutant (N∆34) where 34 of the total N-terminal domains were removed [67]. TEM images revealed the coexistence of empty and AuNP-encapsulating VLPs with a significant size variability. These VLPs were assembled at conditions where no CCMV was formed in the presence of RNA. Purification of the VLPs yielded homogeneous particles of ~20 nm (pseudo *T* = 2), which was supported by a negative-stain TEM reconstruction. Liu et al. compared the effect of AuNP size and surface functionalization on the CCMV VLP formation [68]. Three functional coating agents (bis-p-(sufonatophenyl)phenyl phosphine (BSPP), citric acid and tannic acid) were used to modify AuNPs (7, 12, or 17 nm), respectively. VLPs templated by 7 nm AuNP had a diameter of 18 nm (*T* = 1) and the highest encapsulation efficiency (>90%), while those templated by 12 nm AuNP yielded VLPs of ~23 nm (pseudo *T* = 2) with the lowest efficiency (>50%). The clear difference in the encapsulation efficiency was attributed to the instability of the pseudo *T* = 2 particles. As for the effect of the coating agent, BSPP exhibited the best performance in triggering the VLP formation: more than 99% of the VLPs were successfully loaded with AuNP.

### 3.3. RCNMV

The assembly of RCNMV is triggered by the specific binding between the virion CPs and an RNA sequence(s) and/or structural element(s), known as the origin of assembly (OAS) [69]. This specific recognition discriminates against the encapsulation of heterologous RNA from the surroundings. Progressive binding leads to the cooperative CP wrapping around the RNA genome and the final formation of the icosahedral virion. Decoration of the OAS element on a substrate surface thus provides a feasible method for the encapsulation of noncognate cargos into RCNMV capsids.

In Franzen’s group, the encapsulation ability of various NPs into RCNMV VLPs has been evaluated (Figure 5) [70,71]. A synthetic OAS was first tethered to the NP surface and a subsequent addition of RCNMV CPs yielded the NP encapsulation, as illustrated in Figure 5. The initial report demonstrated a successful loading of 5 nm or 15 nm AuNPs, producing 33.5 ± 3 nm VLPs (*T* = 3) [70]. No VLP formation was observed with 20 nm AuNPs, which was attributed to the constrained size of the capsid cavity (17 nm). Further study using magnetic NPs and QDs unveiled that the RCNMV VLP formation was independent of the core composition and size, provided that the size was smaller than the cavity [71]. NPs with sizes ranging from 4 nm to 15 nm produced uniform VLPs with an average diameter of 32.8 nm (*T* = 3), which was distinct from the CCMV VLP that was highly dependent on the template size [68]. RCNMV VLPs have also been used to load a chemotherapy agent, doxorubicin, through electrostatic interaction [72]. Additional functionalization on the capsid surface with targeting peptides resulted in a bi-functional system, and a selective cytotoxic effect on cancer cells was demonstrated, showing potential applications in diagnosis and therapeutics [73].

### 3.4. AaLS

Recombinant protein expression in *Escherichia coli* produces riboflavin synthase (AaRS)-encapsulating AaLS [9,74]. A subsequent study revealed that the specific encapsulation is mediated by a C-terminal peptide (12 amino acids) on AaRS [74]. Fusion of this particular peptide to heterologous molecules leads to a successful guest compartmentation into the AaLS host [74]. Site-specific mutation generates AaLS cages with an interior net negative charge, and the efficient heterologous encapsulation through electrostatic interaction has been demonstrated [9].

Initially, the Hilvert’s group mutated four residues per monomer to glutamates, yielding an anionic environment inside the capsid (AaLS-neg) [75]. Coproduction of AaLS-neg with a deca-arginine tag (R10) containing green fluorescent protein (GFP) produced a proteinaceous complex through complementary electrostatic interaction. Compared to the AaLS-wt (18 nm, *T* = 1), a size increase to 29 nm (*T* = 3) was observed, a consequence in line with the reduced thermal stability. Since the electrostatic interaction is the dominating force in the sequestration, the possible packaging of any tagged payload (such as human immunodeficiency viruses (HIV) protease) into the mutated capsid can be anticipated (Figure 6a) [76]. Directed evolution produced a variant, AaLS-13, with an improved charge density after four rounds of mutagenesis and selection. TEM results revealed an icosahedral symmetry with a size of ~35 nm (*T* = 3 or *T* = 4). The sequestration of the tagged HIV protease was found to be 5 to 10 times higher than that in the AaLS-neg. Later, they used this highly selected variant as a nanocontainer for efficient substrate sequestration through electrostatic interaction, mimicking natural organelles. The initial study revealed the exceptional loading capacity of a supercharged GFP, GFP(+36) (Figure 6b) [77]. The guest loading can be achieved before or after AaLS-13 cage formation and as high as 45 GFP(+36) molecules per capsid were achieved [78]. Given the supercharged and fluorescent nature of GFP(+36), it was further fused to selected enzymes to facilitate and monitor the enzyme loading in AaLS-13 [78,79,80]. Frey et al. achieved the encapsulation of an enzyme cascade in the AaLS capsid with a local enzyme density of up to 5 mM [79]. The protein shell protected the enzymes against proteolysis but showed limited catalytic enhancement. Azuma et al. demonstrated the robustness and universality of this method by packaging eight active enzymes in AaLS-13 separately [78]. They also created a proteasome-mimicking nanoreactor by the GFP(+36)-aided diffusion of a Tobacco Etch virus protease into AaLS-13, achieving a selective hydrolysis of cationic peptides over zwitterionic or negatively charged peptides [80]. The same cage also successfully encapsulated engineered ferritin cages that possessed a positively charged surface, forming a Matryoshka-type complex [81]. Aside from the intact cage, AaLS-13 pentamers were found to effectively bind positively charged AuNPs through electrostatic interactions, producing a core-shell structure with enhanced colloidal stability over a wide range of pH and ionic strength [82].

The AaLS-based charge complementary system can be reversed through the introduction of positively charged mutations or arginine-rich motifs in the lumen. In Woycechowsky’s group, this was realized by selective mutation on each subunit of wild-type AaLS with four positive amino acids, producing a net charge of +240 in the lumen [83]. The engineered capsid was assembled during the production and simultaneously encapsulated RNA through electrostatic interactions. Encapsulation selectivity over the RNA length was also observed. Later, the effect of RNA sequence on the encapsulation efficiency was evaluated [84]. It was found that RNA with a flexible structure, which was determined by the sequence, maximized charge–charge interactions with the capsid interior, leading to an encapsulation yield ~200 times higher than the RNA with a rigid shape. However, high encapsulation yields confined the expression levels of the encapsulated mRNA, which may hinder the gene delivery application. Positively charged AaLS variants were obtained in Hilvert’s group by co-assembling circularly permuted AaLS that contained arginine-rich motifs with wild-type AaLS [85]. Endogenous RNA was captured during the assembly process and the encapsulated RNA size could be controlled by changing the number and length of the attached arginine-rich motifs. Particularly, short-lived RNA species can be trapped in the capsid, demonstrating the potential in RNA sampling technology. Through an evolutionary optimization, the circularly permuted AaLS capsid proteins were able to recognize and package its encoding RNA, showing a similar behavior as viral protein cages [86,87]. Such findings may provide meaningful insights into virion assembly and evolution.

### 3.5. Other Protein Cages

Many other protein cages have also been employed to encapsulate heterologous cargos through electrostatic interactions. Inherent heat and chemical stability have made ferritin one of the most intensively studied protein cages in the biomedical field and bionanotechnology [6]. The *Archaeoglobus fulgidus* ferritin (AfFtn), a microbial ferritin, presents an outstanding example in the encapsulation of positively charged cargos, such as colloidal gold and GFP(+36) [88,89,90]. Thermosome has a barrel-shaped structure with an outer diameter of 16 nm and a cavity size of 7 nm [91]. After chemical modification in the cavity with a polycationic dendrimer, poly(amidoamine), anionic AuNP and interfering RNA were successfully located in the chamber, and successful RNA interference in cells was induced [92,93]. Vault particles (VP) are ribonucleoprotein particles that are conserved in most higher eukaryotes and have a hollow cylindrical capped barrel morphology with a diameter of 41 nm and a length of 73 nm [94,95,96]. Successful loading of MPS-PPV into VPs through complementary electrostatic interaction was achieved by Tlobert et al. [97]. Another fascinating cage family is the bacteriophage VLPs, which have been shown to package heterologous enzymes with preserved activity and stability [98,99,100]. Recently, computationally designed and evolutionally optimized approaches have emerged as powerful tools to engineer synthetic capsids [101,102,103]. Compared to their native counterparts, the engineered nucleocapsids may circumvent safety risks, showing attractive potential in biomedical fields.

## 4. Virus-Inspired Core-Shell Structure Formed by Protein (Shell) and Polyelectrolyte (Core)

Just like other inspirations acquired from nature, naturally occurring protein cages motivate research in the construction of novel protein cage-mimicking structures for applications in materials science and therapeutics [104]. Selective sequestration of polyelectrolytes into a protein capsule may protect the payload, circumvent the intrinsic toxicity and even enhance the targeted cell uptake [105,106]. While viral capsids serve as a versatile platform, genetically and chemically engineered proteins may circumvent potential immune responses in the host, thus playing an indispensable role in medical applications [107,108]. Moreover, thanks to the tremendous development in protein engineering, artificial proteins can nowadays be produced on a large scale yet with arbitrary structure design and precise size control [109]. In this section, we briefly summarize advances in the development of the electrostatic-driven polyelectrolyte encapsulation in protein shells based on the entrapped cargo species.

### 4.1. Nucleic Acids

As the genetic information carrier, nucleic acid also serves a leading role in diagnosis and gene therapy [110]. However, inherent liability to chemical and enzymatic degradation requires additional functionality to withstand complex conditions in vitro and in vivo. Vazquez et al. fused a poly-arginine (R9) and a poly-histidine (H6) to the termini of a GFP (R9-GFP-H6), which self-assembled into a stable 20 × 3 nm disk-shaped NP [111]. Mixing R9-GFP-H6 with a plasmid DNA produced a cage-like structure with a protein shell and a DNA core. Efficient penetration and transgene expression in the cell nucleus were observed, which was believed to originate from the positive nature and the unique disk-like structure of the protein conjugate. Subsequent structural study on the R9-GFP-H6-DNA complex revealed the coexistence of spherical and tubular nanostructures with plasmid DNA located in the core [112]. Hernandez-Garcia et al. designed a protein copolymer containing a bulk hydrophilic block and a small DNA-binding block [113]. Unlike R9-GFP-H6, individual DNA duplex was wrapped by protein copolymers without aggregation and the strand-like shape was retained. Later, they changed the nonspecific binding block to a DNA-specific binding domain, Sso7d from the hyperthermophilic archaebacterium *Sulfolobus solfataricus* [114]. It was found that the protein copolymer could protect a library of DNA structures (Figure 7a). The DNA stability against DNase and thermal denaturation was also enhanced. Inspired by native viruses, they also designed and produced a triblock protein copolymer [19]. Aside from the hydrophilic block and the cationic binding block, an interlocking block was introduced in the middle, which provided an interprotein interaction. Elucidation of the binding between the protein copolymer and a linear DNA template revealed that the assembly kinetics was similar to an existing model for TMV assembly. Moreover, the protein copolymer proved to be effective in protecting DNA against enzymatic digestion and in transfecting cells. In contrast to the biological means for producing DNA-binding proteins, Kostiainen et al. modulated natural proteins through chemical modification with dendrimers [115]. They functionalized a bovine serum albumin (BSA) and a genetically engineered cysteine mutant of Class II hydrophobin with maleimide-containing dendrimers in a one-to-one manner [116,117]. The obtained protein–dendrimer conjugates exhibited notable DNA binding efficiency, which presented a promising biocompatible vector for gene transfection [117]. Recently, the protein–dendrimer (BSA-G2) was used to coat a DNA origami (60 helix-bundle, 60HB), which was further employed for cell transfection studies (Figure 7b) [118]. Results showed that 60HB was covered with the conjugates in an isolated form, and the stability of 60HB against endonucleases was significantly improved. Furthermore, enhanced cell transfection was observed while the activation of the interleukin 6-mediated immune response was attenuated.

### 4.2. Nanoparticles

Compared to polymeric materials, inorganic NPs possess distinct properties and hold great potential in imaging and sensing [119,120]. Surface-coating of NPs with a protein corona not only offers a robust way to boost the performance, but also helps researchers to understand and predict their fate in living systems. It is generally accepted that uncoated NPs are in a dynamic exchange with proteins and other biomolecules in a biological milieu [121,122]. Many studies have unveiled the dynamic interactions using various characterization techniques [123,124,125,126,127]. Meanwhile, great efforts have also been devoted to improving the medical performance of NPs by protein coating. Simak et al. compared the influence of protein coatings on the biocompatibility of carboxylated carbon nanotubes (CNTCOOH), including albumin (HSA), fibrinogen (FBG), γ-globulins (IgG) and histone H1 (H1) [128]. The results revealed that proteins with a high positive charge (H1), small size (HSA) or high conformational flexibility (HSA) were more inclined to bind CNTCOOH. The effect of the protein corona on human blood platelets (PLT) varied: while HSA and FBG prevented the aggregating activity of CNTCOOH, H1 corona triggered serious PLT aggregation, which was similar to bare CNTCOOH, implying the risk of thrombogenesis. Treuel et al. compared the coating efficiency of modified HSA on dihydrolipoic acid-coated quantum dots (DHLA-QD), and the cell uptake performance of the coated DHLA-QD was evaluated [129]. They found that the protein binding was fully reversible. Compared to native HSA, ethylenediamine-modified HSA (HSAam) exhibited significantly high binding affinity to DHLA-QD (1000×) while succinic anhydride functionalization (HSAsuc) showed slightly decreased binding affinity (1/3×) (Figure 8a). They also found that the cationic modification (HSAam) increased plasma membrane binding, but subsequent cell internalization was slow. Ma et al. used dissipative particle dynamics simulation to investigate the binding between HSA and NP with different surface properties, and the subsequent in vivo transportation was compared [106]. They found that positive NPs exhibited better HSA-adsorbing ability than hydrophobic ones (Figure 8b). Besides, the protein modification could raise phagocytosis, showing an obvious impact on the immune response. However, the HSA corona lowered the cellular uptake efficiency in targeted tumor tissues. Zhou et al. found that cell uptake of BSA-coated graphene oxide (GO) was reduced and the resulting cytotoxicity of GO was mitigated, a similar phenomenon that was observed with spherical NPs and supported by the molecular dynamics simulation [130,131].

### 4.3. Organic Polymers

Duracher et al. synthesized core-shell latex particles containing a polystyrene (PS) core and a positively charged poly(N-isopropylacrylamide) (pNIPAM)) shell, which was used to absorb and release proteins [132]. BSA was used as a study model. It was found that the protein adsorption efficiency was influenced by temperature, pH and salinity, and the desorption efficiency was dependent on the adsorption rate. Temperature change across the lower critical solution temperature of the pNIPAM layer dramatically changed the surface charge density on PS and further affected the BSA binding and release efficiency, a property that is desirable for controlled release. Yan et al. designed a similar system containing a magnetite nanocrystal cluster core and a poly(2-(dimethylamino) ethyl methacrylate) hairy shell [133]. The obtained complex was magnetic-, temperature- and pH-responsive. Excellent BSA-binding ability (maximum 660 mg BSA per 1 g complex in 5 min) was achieved, and the quantitative desorption could be realized by lowering pH or increasing salinity concentration. These unique properties were anticipated for protein separation and purification. Gref et al. investigated the plasma protein adsorption ability of PEG-coated poly(lactic acid) (PLA), poly(lactic-co-glycolic acid) (PLGA) and poly(ε-caprolactone) (PCL) NPs to predict their fate in vivo and further improve their performance as a drug delivery platform [134]. It was found that plasma proteins were absorbed to the anionic core of the NPs, and that the process was significantly affected by the PEG size and coverage density. Gossmann et al. compared the serum protein absorption performance of two oppositely charged polymeric NPs: positively charged PLGA NPs stabilized with didodecyldimethylammonium bromide and negatively charged HSA [135]. Approximately five-fold more proteins were detected on the PLGA surface and the protein diversity was much higher than that on negative HSA NPs.

## 5. Conclusions

This review highlights four native protein cages and their utilization in noncognate polyelectrolyte encapsulation. Following that, the protein cage-mimicking structures that were assembled through electrostatic interactions containing a polyelectrolyte core and a protein shell were discussed. Undoubtedly, polyelectrolytes offer a promising solution for many challenges in therapeutics and bioimaging [136,137]. However, the intrinsic liability to chemical or enzymatic degradation (nucleic acids) and the off-target accumulation and toxicity (inorganic NPs) are the major obstacles [20,110]. Sequestration of polyelectrolytes into a protein cage presents a promising approach in resolving the dilemma owing to their unique features. First, the cationic nature of the inner cavity and interprotein interactions allow for an efficient polyelectrolyte loading and a stable cage formation, where the payload is preserved and protected. Second, the limited number of constitutional capsid subunits eases the modification efforts, because a single modification on one subunit can lead to multiple functional sites on the cage surface. Combinational functionalization or the encapsulation of materials that are inaccessible to native cages thus can be envisioned. For example, PSS was reported to stabilize CCMV capsids that were coated with PEG chains, a functionalization that provoked irreversible dissociation of the virus [54]. Such functionalization is anticipated to prolong the circulation time and function as a stealth coating to decrease immune response in the body [138]. Third, computation-aided de novo design offers the possibility of constructing novel proteins with predetermined structures and functionalities [139]. Last, proteins generally display limited cytotoxicity compared to plain polyelectrolytes, and selected protein capsids can even target and accumulate in tumor cells [140].

We have outlined how the polyelectrolyte encapsulation within protein cages not only offers a robust method for confinement, but also serves as an intriguing method to modulate how polyelectrolytes interface with biology. Taken together, these developments highlight the potential of the protein cage-mimicking structures in diverse applications. However, most reports have concentrated on proteins as capsules with little attention to their tailored functionalization and stimulus-responsiveness of both the encapsulated polyelectrolyte and the protein shell. Therefore, efforts on addressing these aspects are imperative to realize the full potential of this multidisciplinary research field.

## Figures and Tables

**Figure 1 pharmaceutics-13-01551-f001:**
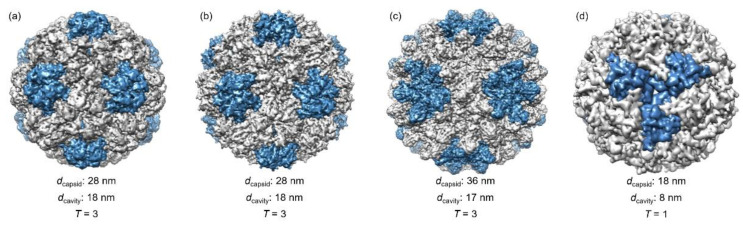
Structures of protein cages highlighted in this review (not drawn to scale, blue color indicates the 12 pentamers and gray color indicates the 20 hexamers in (**a**–**c**)): (**a**) Brome mosaic virus (BMV) (PDB 1JS9). (**b**) Cowpea chlorotic mottle virus (CCMV) (PDB 1CWP). (**c**) Red clover necrotic mosaic virus (RCNMV) (PDB 6MRM) and (**d**) lumazine synthase from Aquifex aeolicus (AaLS) (PDB 1HQK). Blue color indicates three protein subunits.

**Figure 2 pharmaceutics-13-01551-f002:**
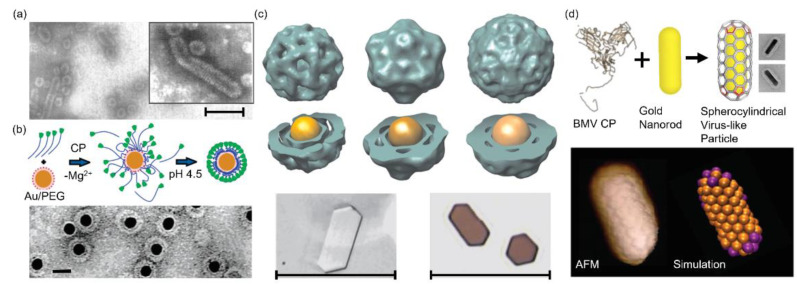
BMV VLPs assembled around diverse templates. (**a**) TEM images of BMV rod-like particles templated by a calf thymus dsDNA. Scale bar: 100 nm. Reprinted from [33]. Copyright (1969) with permission from Elsevier. (**b**) Top panel: The proposed BMV VLP formation templated by carboxylate-modified AuNPs. Bottom panel: A representative TEM image of BMV VLPs encapsulating 12 nm AuNPs. Scale bar: 20 nm. Reprinted with permission from [35]. Copyright (2006) American Chemical Society. (**c**) Top and middle panel: 3D reconstructions of BMV VLPs templated by AuNPs of different sizes (from left to right: 6, 9 and 12 nm). Bottom panel: transmission optical images of (left) R3BMV crystals and (right) VLP_12_ crystals. Scale bar: 100 μm. Reprinted with permission from [36]. Copyright (2007) National Academy of Sciences. (**d**) Top panel: Schematic illustration and TEM images of BMV VLPs formed around gold nanorods. Bottom panel: The structure imaged by AFM and the corresponding simulated structure. Reprinted with permission from [37]. Copyright (2018) American Chemical Society.

**Figure 3 pharmaceutics-13-01551-f003:**
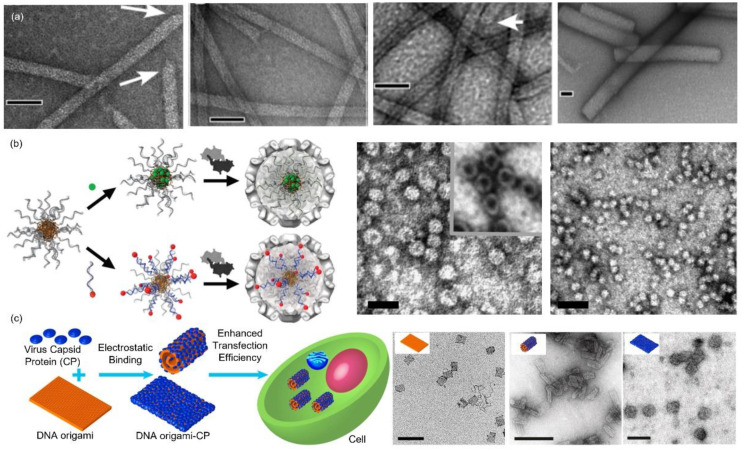
CCMV VLPs templated by DNA nanostructures. (**a**) Representative TEM images of CCMV capsids templated by a 500 bp dsDNA with bp:CP dimer ratios of (from left to right): 1:1, 7:1, 14:1 and 28:1. White arrows mark the hemispherical caps. Scale bar: 50 nm. Reprinted with permission from [48]. Copyright (2006) American Chemical Society. (**b**) Left panel: Schematic illustration of CCMV capsid assemblies templated by DNA micelles and the proposed approaches for drug loading. Right panel: Representative TEM images of (left) assembled CCMV VLPs and (right) free DNA micelles. Scale bar: 40 nm. Reprinted with permission from [50]. Copyright (2010) American Chemical Society. (**c**) Left panel: Schematic illustration of CCMV CP coated DNA origami and the subsequent cellular delivery. Right panel: TEM images and the corresponding illustrative models (insets) of (left) the original rectangular DNA origami, (middle) the rolled tubes of CCMV CP coated DNA origami and (right) unwrapped CCMV CP coated DNA origami. Scale bar: 200 nm. Reprinted with permission from [53]. Copyright (2014) American Chemical Society.

**Figure 4 pharmaceutics-13-01551-f004:**
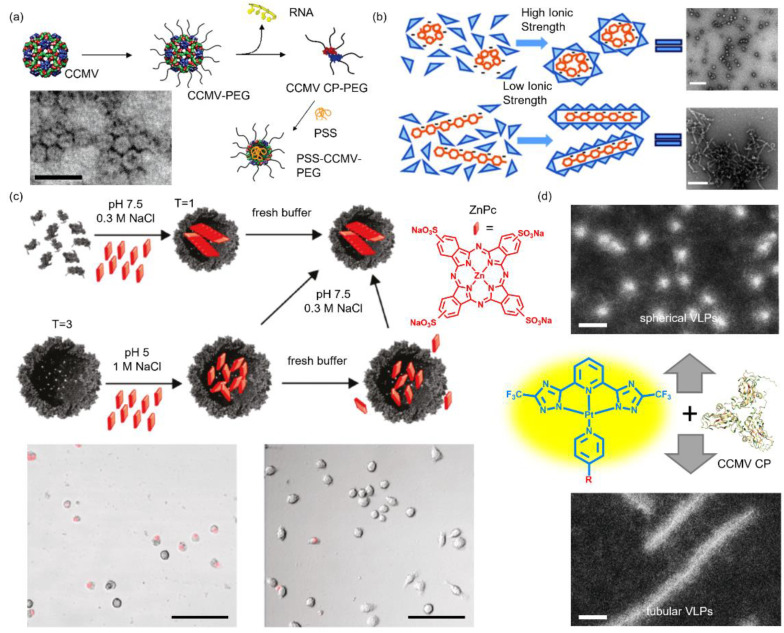
CCMV VLPs assembled around synthetic polymers and amphiphiles. (**a**) Schematic pathway of CCMV-PEG assembly templated by PSS and the characterization result from TEM measurement. Scale bar: 50 nm. Reprinted with permission from [54]. Copyright (2009) American Chemical Society. (**b**) Schematic presentation of the morphological manipulation of CCMV VLPs through the template structure transition that is controlled by NaCl concentration and the corresponding TEM images. Scale bar: 200 nm. Reprinted with permission from [60]. Copyright (2011) American Chemical Society. (**c**) Schematic illustration of the encapsulation of ZnPc dyes into CCMV VLPs and the cell images (bottom left panel) in the presence and (bottom right panel) in the absence of ZnPc-loaded capsids. Both images are the overlap of fluorescence images and transmission images of RAW 264.7 macrophage cells. Scale bar: 100 μm. Reprinted with permission from [62]. Copyright (2011) American Chemical Society. (**d**) Spherical and tubular CCMV VLPs templated by tailor-made Pt(II) amphiphiles. Scale bar: 50 nm. Reprinted with permission from [63]. Copyright (2018) American Chemical Society.

**Figure 5 pharmaceutics-13-01551-f005:**
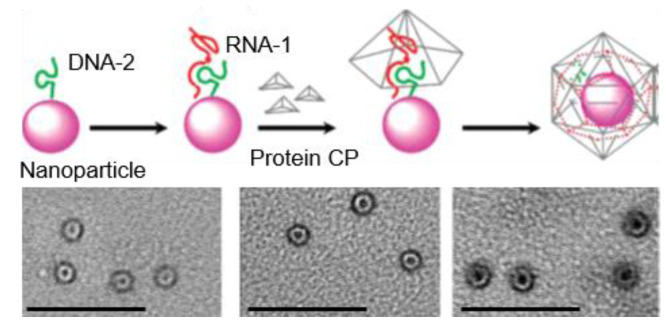
RCNMV VLPs templated by NPs. (**Top panel**): schematic representation of the templated assembly of RCNMV on NPs. (**Bottom panel**): TEM images of RCNMV VLPs containing magnetic CoFe_2_O_4_ NPs of different sizes. Scale bar: 200 nm. Reprinted with permission from [71]. Copyright (2007) American Chemical Society.

**Figure 6 pharmaceutics-13-01551-f006:**
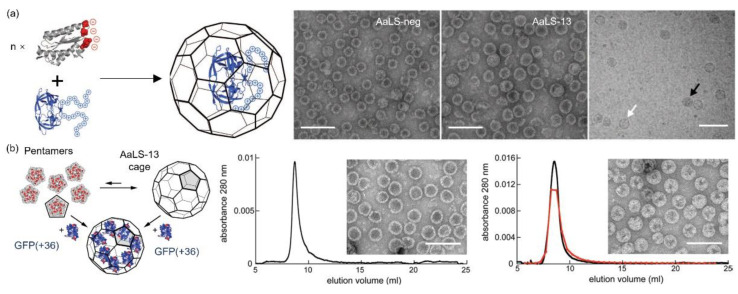
AaLS mutants and the sequestration of cationic cargos. (**a**) Schematic illustration of the loading of a R10-tagged HIV protease into the non-quasi-equivalent AaLS variant, AaLS-13, and the EM images of (left) AaLS-neg, (middle) AaLS-13 and (right) AaLS-13 coproduced with HIV protease-R10 (white and black arrows indicate empty and protease-loaded cages, respectively). Scale bar: 100 nm. From [76]. Reprinted with permission from AAAS. (**b**) Schematic illustration of the two approaches to load AaLS-13 with GFP(+36), and the characterization results (chromatogram and TEM) of (left) empty and (right) loaded AaLS-13. In both chromatogram graphs, black lines indicate protein absorption and red dots indicate the relative fluorescence of individual fractions. Scale bar: 100 nm. Reprinted with permission from [77]. Copyright (2012) American Chemical Society.

**Figure 7 pharmaceutics-13-01551-f007:**
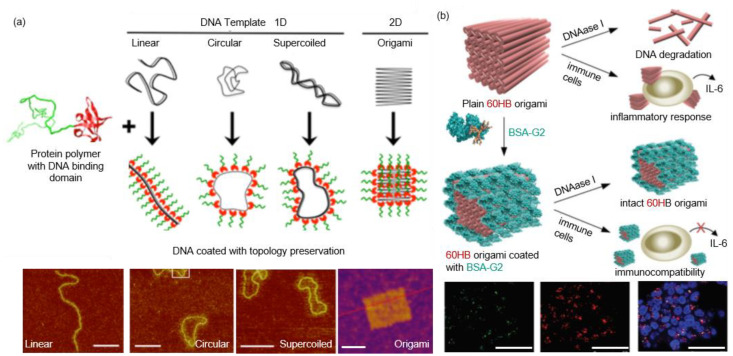
DNA nanostructures coated with proteins through electrostatic interactions. (**a**) Top panel: Schematic representation of DNA nanostructures coated with protein polymers. Bottom panel: Corresponding AFM images of DNA nanostructures coated with protein polymer. Scale bar: 50 nm. Reprinted with permission from [114]. Copyright (2017) American Chemical Society. (**b**) Top panel: Schematic illustration of the 60HB origami coated with BSA-G2 and the subsequent shielding effect against DNAase. Bottom panel: Confocal images of HEK293 cells after transfection with BSA-G2-coated 60HB for 12 h. The left panel is the green AlexaFluor 488 of origami channel, the middle panel is red LysoTracker, and the right panel contains the overlap of the left and middle panel and DAPI-stained nuclei. Scale bar: 50 μm. Reproduced with permission from [118].

**Figure 8 pharmaceutics-13-01551-f008:**
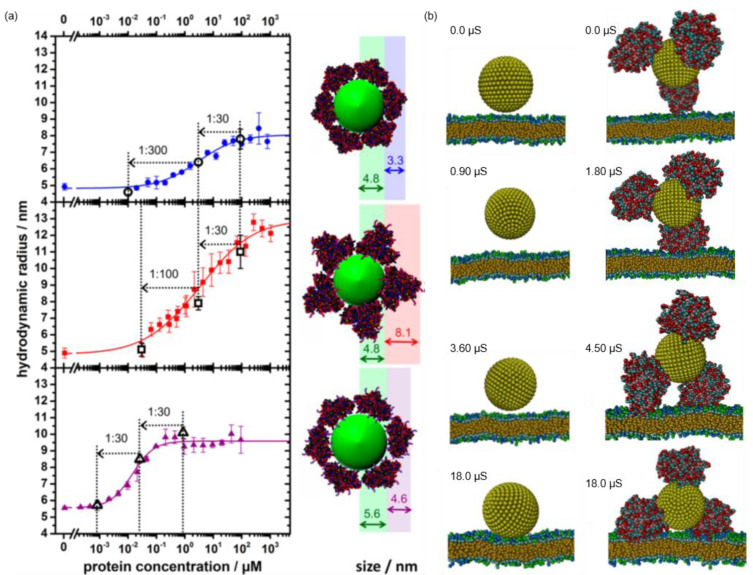
Experimental and simulated demonstrations of protein coated NPs. (**a**) Adsorption of (top) plain HSA, (middle) HSAsuc and (bottom) HSAam onto DHLA-QDs as a function of protein concentrations and the corresponding schematic depictions of the hydrodynamic radii increase as a result of the protein adsorption. Reprinted with permission from [129]. Copyright (2014) American Chemical Society. (**b**) Simulated time sequence of the snapshots of the interactions between macrophage cell membranes and positively charged NPs. Left panel: in the absence of serum proteins. Right panel: in the presence of serum proteins. Reproduced with permission from [130].

## Data Availability

Not applicable.

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
