# Peer review of "Polyelectrolyte Encapsulation and Confinement within Protein Cage-Inspired Nanocompartments"

_pharmaceutics, 2021, doi:10.3390/pharmaceutics13101551_

Round 1

Reviewer 1 Report

In this review article, Lin et al. describe how electrostatic interaction has been exploited for packaging a variety of foreign guests such as metal nanoparticles, polymers, and proteins in proteinaceous compartments. They particularly focus on four protein cage systems, three viral capsid and one cage-forming lumazine synthase among others. Such general and efficient guest encapsulation systems can be bases for a wide variety of medical and nanotechnological applications, and thus I believe that this review will attract a great attention from a broad segment of the MDPI Pharmaceutics readers. The review covers previously reported viral capsid systems quite well whereas there are some misunderstanding of AaLS-based system as well as lacks of some important publications. The manuscript reads well in general, yet it contains some minor inappropriate scholarly presentations. The authors should address the points listed below;

The studies the author should consider to add;

  1. Artificial protein cages: Naturally occurring protein cages have inspired design and construction of artificial equivalents, which is currently a very hot scientific topic (Stupka I and Heddle JG., Curr. Opin. Struct. Biol. 64, 66-73 (2020)). Indeed, the authors also describe this point in the conclusion section, line 630, “Third, computation-aided de novo design offers the possibility for constructing novel proteins with predetermined structures and functionalities [114].” Some of the computationally designed protein cages were further engineered and evolved to package guest molecules using electrostatic interactions, Butterfield GL, et al. Nature, 552, 415–420 (2017); Edwardson TGW, et al. J. Am. Chem. Soc., 140, 10439–10442 (2018); Edwardson TGW, et al. Nat. Commun., 11, 5410 (2020). Although this review mainly focuses on the four protein cage systems, these studies on artificial protein cages will significantly increase the reader’s attention and thus can be added in page 12, section 3.5. Other protein cages. In the same section, the authors may also consider to include the bacteriophage Qb system developed in the MG Finn’s group, Fiedler JD, et al., Angew. Chem. Int. Ed. 49, 9648–9651(2010); Das S, et al., Biochemistry 59, 2870–2881 (2020), as their system is one of the pioneering works in the field.
  2. Other AaLS-based systems: Like the viral capsid systems, the engineered AaLS-13 cage has been demonstrated to package metal nanoparticles via electrostatic interaction, Sasaki E., et al, ChemBioChem, 21, 74-79 (2020) (This study concept is more close to the 4. Virus-inspired core-shell structure formed by protein (shell) and polyelectrolyte (core)). The same cage also successfully encapsulated engineered ferritin cages possessing a positively supercharged surface, Beck T, et al. Angew. Chem. Int. Ed., 54, 937-940 (2015). Moreover, it has also been shown that AaLS-based charge complimentary system can be reversed: introduction of positively charged mutations or arginine-rich motifs in the lumen to package negatively charged guest molecules, Lilavivat S, et al. J. Am. Chem. Soc., 134, 13152-13155 (2012); Fu J and Woycechowsky KJ, Biochemistry, 59, 1517–1526 (2020); Azuma Y. et al, J. Am. Chem. Soc., 140, 566–569 (2018); Terasaka N, et al. Proc. Natl. Acad. Soc. U.S.A., 115, 5432-5437 (2018); Tetter S., et al., Science, 372, 1220-1224 (2021). Since AaLS is one of the main systems highlighted in this article, these studies should be included in the section 3.4. AaLS.
  3. Triangular number and cryoEM study of the engineered AaLS cages: This quasi-equivalent definition of protein cages can be confirmed by a reasonably high resolution structural analysis such as X-ray crystallography or single particle reconstruction of electron microscopic images. All other methodologies such as simple TEM imaging, DLS, or SAXS can be used to approximate the assembly status, but cannot confirm it. For example, the assemblies of the AaLS variant possessing four negatively charged mutations, AaLS-neg, and the further evolved variant AaLS-13 were predicted to be T=3 and T=3/4 state cages, respectively (page 11, line 460 and page 12, line 465). But, the latter cryoEM single particle reconstruction study revealed that the proteins form non-quasi-equivalent assemblies composed of 180- and 360-copies of proteins, Sasaki E., et al. Nat. Commun., 8, 14663 (2017). In this regard, Page 1, line 38, “Lumazine synthase from Aquifex aeolicus, on the other hand, is built from 60 or 180 identical subunits that assemble into icosahedral capsids with outer diameters of 16 nm or 30 nm [6,7].” The references 6 and 7 do not confirm any 180-mer assembly. Expanded 180-mer AaLS assemblies were observed only for engineered variants, above mentioned AaLS-neg and AaLS-IDEA, by cryoEM single particle reconstruction, Sasaki E., et al. Nat. Commun., 8, 14663 (2017); Ladenstein R., et al., FEBS J. 280, 2537-2563 (2013). But, I would not include this information about engineered variants in this introduction section as I assume the authors want to describe the wildtype AaLS assembly here. Furthermore, authors should explain briefly about the Caspar and Klug theory in the ‘Native structure of protein cages’ section.

Inappropriate scholarly presentations:

  1. Line 20, “Selective encapsulation can protect the payload from the surroundings, shield the potential toxicity or even enhance the intended performance of the payload, which is intriguing in drug- or gene-delivery and imaging.”

“intriguing” may be rephrased like “attractive” or “appealing”.

  1. Line 27, “While sub-cellular compartments developed in eukaryotes are comprised of lipids, likewise in prokaryotes, micro- or nano-compartments are dominantly protein-based [1].”

The reference 1 is exclusively for bacterial microcompartments. The authors should also cite references for bacterial nanocompartments, e.g. encapsulin.

  1. Line 29, “In either case, precise compartmentalization can sustain localized and concentrated sequentially acting enzymes and substrates, which boosts the rate of catalytic cascades or, in many cases, sequesters the produced toxic substances [1–3].”

Sequestering toxic intermediate is the case for Pdu/Eut compartments. What else? Is it really in “many cases”? The sentence is also difficult to read, especially “sustain localized and concentrated sequentially acting enzymes and substrates”.  

  1. Line 33, “Protein cages, such as the cowpea chlorotic mottle virus (CCMV) and ferritin, are the simplest forms of nanocompartments found in nature”.

I would not call the CCMV capsid “the simplest” since the protein assembly mechanism is not yet fully understood and it has shown polymorphological behavior. This should be rephrased like “simple forms”? Also, CCMV is a virus but not a protein cage.

  1. Line 36. “For example, mammalian ferritins consist of 24 copies of subunits that construct a spherical structure with an outer diameter of 12 nm and an inner cavity of 8 nm [5].”

This sentence may be more precise. Mammalian ferritins do not always consist of 24 identical subunits. The authors need to be specific as described in the review, Chakraborti S, Chakrabarti P (2019) Adv. Exp. Med. Biol., 1174, 313–329 (2019).

  1. Line 41. “While the ferritin regulates the availability of iron in almost all organisms, lumazine synthase is found specifically in bacteria and responsible for the production of riboflavin (vitamin B2) [5,8].”

This is not true. Lumazine synthase is found in fungi, yeasts, plants, archaea, and eubacteria, Ladenstein R., et al., FEBS J. 280, 2537-2563 (2013).

  1. Lane 43, “Viral protein cages are typically loaded with nucleic acid genomes”

Viral protein cages are loaded with nucleic acid genomes. Why “typically”?

  1. Lanes 45, “…target and enter hosts and release the genetic materials in response to external stimulus”

Do all the viral cages enter host cells? What is external stimulus? This sentence can be more generalized such as “deliver genetic materials to host cells”

  1. Line 49, “Given the polyanionic nature of genetic materials, it is expectable to anticipate the successful capsid formation templated by other polyanions.”

“expectable to anticipate” is redundant. Also, this is the capsid formation mechanism of small RNA viruses. Coat proteins from large DNA viruses assemble first and then take up genomic information using a pump.

  1. Line 73‘…Bacteriophage P22’

There is no need for a capital letter in the “bacteriophage”.

  1. Line 79, “In contrast to non-viral protein cages which generally exist with a spherical shape and a small cavity, viral protein cages adopt various morphologies and have cavity diameters ranging from ~30 nm (bacteriophage FX174) to over 500 nm (mimivirus)”.

I disagree with the sentence. Non-viral protein cages are diverse in terms of size, shape, and functions. Examples of such cages with non-spherical shape or large size include bacterial microcompartments (BMCs) such as carboxysome and Pdu/Eut compartments (submicrometer), ATP-dependent proteasome and the bacterial counterparts ClpAP/XP (cylinder), and Vault (ovoid, 70 x 40 nm) which is actually mentioned in this manuscript (line 491).  

  1. Line 93.

PDB entry for the BMV capsid should be 1JS9. The colors in the figure should be explained in the legend.

  1. Line 111, “It structurally resembles BMV, containing an icosahedral capsid (T = 3) that packages a single-stranded RNA (ssRNA) genome with an 112 outer diameter of 28 nm and an inner cavity diameter of 18 nm (Figure 1b) [16].”

To keep the consistency to BMV, it is better to include the information that CCMV is also a (+)ssRNA virus. The word “containing” can cause a misreading. The author may describe like “Like BMV, the coat protein forms icosahedral capsid…”

  1. Line 134, “…lumazine synthase from Aquifex aeolicus (AaLS) is found to self-assemble into a cage-like structure, which consists of 60 copies of identical subunits and adopts an icosahedral symmetry with an outer diameter of 18 nm (T = 1, Figure 1 d) [6]”

The size and assembly status in this sentence conflict to what is described in line 38, “Lumazine synthase from Aquifex aeolicus, on the other hand, is built from 60 or 180 identical subunits that assemble into icosahedral capsids with outer diameters of 16 nm or 30 nm [6,7].”

  1. Lane 179, “in the absence nucleating agents”

In the absence “of” nucleating agents.

  1. Lines 194, “…highly resembling the native R3BMV.”

The “R3BMV” should be explained.

  1. Line 222, “…a similar value as a previously report with 18 nm AuNPs [24].”

Similar to a previously reported value or a similar value to the previous report?

  1. Line 227, “Nevertheless, the superparamagnetic IONPs used in this work had a blocking temperature of ~250 K, which makes the IONP-encapsulating VLPs intriguing as magnetic resonance imaging agents and biomagnetic materials.”

Again, “intriguing” can be “attractive/appealing”.

  1. line 284, “Two-enzyme cascade was encapsulated in the VLPs and the system exhibited enhanced catalytic activity, which was attributed to the increased local substrate concentration and the decreased intermediate diffusion pathway.”

Hypotheses and evidenced matters should be distinguished clearly. This is a hypothesis and not confirmed, as the reference 40 states that “the protein concentration determination by gel densitometry is expected to have a large deviation and consequently also the kcat”. It still remains a question either copackaging of sequentially active catalysts in a protein cage can enhance overall reaction rate or not.

  1. Lane 354, “Välimäki et al. conjugated a heparin-specific peptide (HBP) to the CP and the successful CCMV VLP formation was triggered by heparin [49]”

It would be nice if the author could explain the system in more detail.

  1. Line 436, “Wild-type AaLS co-assembles with a riboflavin synthase (AaRS) in the cavity. The AaRS converts the intermediate produced by AaLS, which tremendously enhances the biosynthesis sensitivity and rate [62].”

This sentence is wrong and needs to be more precise. Firstly, the encapsulation was confirmed only in a recombinant form, but never with the native proteins isolated from the hyperthermophile. Secondly, in the recombinant expression experiments (reference 62), one AaLS cage encapsulates three AaRSs (likely a homo trimer) in average, which corresponds to the native enzymes from Bacillus subtilis, Bacher, A., et al. J. Mol. Biol. 187, 75-86, (1986); Ritsert, K., et al. J. Mol. Biol. 253, 151-167, (1995). Thirdly, the catalytic enhancement is known for the Bacillus enzymes, Kis, K. and Bacher, A. J. Biol. Chem. 270, 16788-16795, (1995), but never confirmed with the Aquifex analogous. Finally, the enhancement effect is likely not only by sequestering the intermediate lumazine, but also RS produces the LS substrate as the byproduct whereas the detailed substrate channeling mechanism is yet unknown, Ladenstein R., et al., FEBS J. 280, 2537-2563 (2013).

  1. Line 445, Figure 6 a) and b) showing schematic representation of AaLS cage.

As mentioned in the comment #3, engineered AaLS does not form quasi-equivalent assembly. The authors may include this information in the figure legend since the football-like figure can be misleading. Additionally, the SEC should be explained in the figure legend, e.g. the red dots indicate the green fluorescent signal of each fraction.

  1. Line 454, “Initially, the Hilvert’s group mutated four residues per monomer to glutamates, yielding an anionic environment inside the capsid (AaLS-neg) [63]. AaLS-neg had a lower thermal stability (Tm = 95 °C) than the wild type (AaLS-wt, Tm = 119.9 °C), which was attributed to the destabilization effect caused by the Coulombic repulsion between heterologous glutamates.”

In the reference 63, it was confirmed that AaLS-neg does not denature, in terms of the secondary structure, by heating up to 95 °C. This is not the melting temperature of AaLS-neg and it is yet unknown either AaLS-neg has lower thermal stability than AaLS-wt or not.

  1. Line 485, “…presents an outstanding example in the encapsulation of positive cargos, such as colloidal gold and GFP(+36)”

“positive cargos” should be “positively charged cargoes”.

  1. Line 487, “Thermosome is a group II chaperonin and has a barrel-shaped structure with an outer diameter of 16 nm and a cavity size of 7 nm [72]”

Then, what is the group I chaperonin? Authors should briefly describe what the group II chaperonin defines or delete this information.

  1. Line 523, “Vazquez et al. fused a poly-arginine (R9) and a poly-histidine (H6) to the terminals of a GFP (R9-GFP-H6), which self-assembled into a stable 20 × 3 nm disk-shaped NP [86]”

“Terminals” should be “termini’.

  1. Line 540, “Elucidation on the binding between the protein copolymer and a linear DNA template revealed that the assembly kinetics was similar to an existing model for TMV assembly.”

Elucidation “of” the binding.

  1. Line 563, “Left panel: in the absence serum proteins”

In the absence “of” serum proteins.

  1. Line 547, “The obtained protein-dendrimer conjugates exhibited notable DNA binding efficiency, which presented an excellent biocompatible vector for gene transfection [93]. Recently, the protein-dendrimer (BSA-G2) was used to coat a DNA origami (60 helix-bundle, 60HB), which was further employed for cell transfection studies (Figure 7b) [94]. Results showed that 60HB was covered with the conjugates in an isolated form, and the stability of 60HB against endonucleases was significantly improved. Furthermore, enhanced cell transfection was observed while the activation of the immune response was attenuated, demonstrating the robustness of the BSA-dendrimer conjugate as a delivery vector.

These sentences clearly overclaim the results. I would imagine “an excellent biocompatible vector for gene transfection” with “attenuated immune response” can work in vivo with much higher transfection efficiency than some commonly used virus- and lipid-based vectors without any cytotoxicity and any remarkable immune response. However, in the references 93 and 94, they tested these BSA-dendrimer conjugates only in vitro and only one type of acute immune response readout, and only compared to naked DNAs. I suggest the authors to rephrase these sentences to be more specific and precise.   

Reviewer 2 Report

This is a valuable review summering a huge amount of papers, starting from the structure of cage-shaped proteins to various artificial ways of their protein subunit assembly. The manuscript focused on mainly four cage-shaped proteins, BMV, CCMV, RCNMV, and AaLS from Aquifex aeolicus. The importance of electrostatic interactions which played critical role for the nucleation formation and self-assembly of the cage-shaped shells was well described and its importance was clarified. We could also comprehend the general mechanism of shell construction with this review paper. The manuscript further described the method fabricating filamentous structures using templates. A huge amount of information was well organized. There are many researchers who are considering the application of spherical proteins as carriers of various nanomaterials and for DDS, and I believe this review will be an important guide for such researchers. My few concerns are as follows: 1. Authors describes in INTRODUCTION, ”First, the structures of native protein cages that have been intensively utilized to load extraneous polyelectrolytes are introduced. Next, the assemblies and applications of protein cages encapsulating extraneous polyelectrolytes through electrostatic interaction are addressed. Finally, the protein cage-resembling structures and coatings of proteins that are templated by polyelectrolytes through electrostatic interaction are discussed along the way. Finally, the protein cage-resembling structures and coatings of proteins that are templated by polyelectrolytes through electrostatic interaction are addressed.” And provided two chapters "Protein cages encapsulating noncognate polyelectrolytes" and "Virus-inspired core-shell structure formed by protein (shell) and polyelectrolyte (core)."; These two chapters seemed to be very similar. It would have been easier for the reader to understand if you had summarized each of the four BMVs, CCMV, RCNMV, Aquifex aeolicus-derived AaLS, independently. 2. In the “Native structures of protein cages” chapter, it would be better to have a comparison table of four structures, so that the reader can better understand those structures. 3. line 198 “Such a unique optical property”; I am not sure if this is enough to for readers to understand what is unique. 4. There are a lot of interest in the application of spherical proteins, such as for DDS, but there were not much information on the application in this review. Could you please consider adding a few lines of information on applications for researchers who are considering applications?

Reviewer 3 Report

This review manuscript describes the polyelectrolyte encapsulation within the protein cage.  While this review may contribute to some fields, such as biology, biochemistry, and supramolecular chemistry, I have some reservations that should be addressed before publication.  My comments are following;

  1. As the authors describe in the introduction, there are some excellent reviews concerning this topic. The authors should emphasize why they focus on the advances in the core-shell nanoparticles with a protein shell and polyelectrolyte core.
  2. Most of the data in this review are electron microscopic images. The authors need to show other data and schematic procedures of the respective methods to better understand readers out of the fields.
  3. The “4.3 Organic polymers” section is a poor description compared to other sections. The authors cite only two publications. However, more papers should be cited and may contain the layer-by-layer methods.  The authors should add more publications concerning organic polymers to promote better discussion.

Reviewer 4 Report

This article is a very insightful and well written review of
nanoparticles formed by protein cages around anionic polyelectrolytes.
The review is very current given applications of the formed
nanoparticles for delivery of therapeutics and medical imaging
applications.  Overall, the material covered in the review was very
well selected and looked at the relationship between nanoparticle
structure as a function of the protein and cargo properties, the
assembly/disassembly process, as well as the biological implications.
In addition, a section is included on engineering proteins to forms
nanoparticles along with a discussion of the protein corona.  I don't
have any major comments, and only two minor comments as suggestions,
and a few minor grammatical corrections.

Minor comments:

I would have preferred to see more details for the molecular
(electrostatic) interactions that control the disassembly and assembly
of the particles as a function of ionic strength, pH, and in the
presence of divalent cations.  This knowledge would be transferable
across a range of systems.

In the introduction, the review would be better motivated by
discussing the applications in medical imaging and delivering
therapeutics in more detail, possibly giving specific current
examples.

Superficial corrections:

In Figure 1, would be helpful to define the colour scheme

Line 229: replace serial by series

Round 2

Reviewer 3 Report

The revised manuscript is acceptable.